# Maternal Daidzein Supplementation during Lactation Promotes Growth Performance, Immunity, and Intestinal Health in Neonatal Rabbits

**Hongmei Xie [1,2,3,†], En Yu [1,†], Huamei Wen [2], Bayi Jiang [2,3,*], Guihua Fu [2], Haitao Sun [2] and Jun He [1,*]**

[1] Animal Nutrition Institute, Sichuan Agricultural University, Chengdu 611130, China; hmx861@163.com (H.X.); 2019314044@stu.sicau.edu.cn (E.Y.)

[2] Animal Science and Technology Institute, Shandong Vocational Animal Science and Veterinary College, Weifang 261061, China; whm1126@126.com (H.W.); zgfugh@126.com (G.F.); wwww8888@163.com (H.S.)

[3] Weifang Key Laboratory of Animal Nutritional Physiology, Weifang 261061, China

* Correspondence: sdmyby@163.com (B.J.); hejun8067@163.com (J.H.); Tel.: +86-28-86291781 (J.H.)

† These authors contributed equally to this work.

**Abstract:** The main purpose of the present research was to evaluate the effect of varying levels of DA inclusion in maternal diet, in the form of powder, on the born-weaning growth performance (days 1–30) and intestinal health of neonatal rabbits. A total of 152 delivered maternal does ($3.94 \pm 0.05$ kg) were allocated into four groups, with thirty-eight replicates of one doe each, and fed with a control diet (CON) supplemented with different levels of powdered DA (85 mg/kg (DA85), 170 mg/kg (DA170), and 340 mg/kg (DA340)) during lactation. The results show that dietary DA increased individual body weight at days 21 and 30 ($p = 0.03$ and $p < 0.01$) and showed a linear and quadratic effect on individual body weight and average daily gain (ADG) ($p < 0.05$), although ADG was not affected by DA. The serum progesterone (P) ($p = 0.05$) and glutathione peroxidase (GSH-Px) ($p = 0.01$) concentrations of maternal rabbits were increased in all maternal DA-supplemented groups and showed a linear and quadratic effect ($p < 0.05$), while a linear effect was observed in estrogen (E-2) ($p < 0.05$). Interestingly, the concentrations of the serum insulin-like growth factor-1 (IGF-I) ($p < 0.01$) and immunoglobulin (IgA and IgG) ($p < 0.01$ and $p = 0.01$) in neonatal rabbits were increased in all dietary DA groups, and both showed a linear and quadratic effect ($p < 0.05$). In addition, the serum interleukin (IL-1β) ($p = 0.03$) and catalase (CAT) ($p = 0.04$) concentrations were affected by DA supplements, but linear and quadratic effects were only observed in the catalase (CAT) of neonatal rabbits ($p < 0.05$). Importantly, the duodenal and jejunal villus heights ($p < 0.04$ and $p = 0.01$) and the ratio of villus height to crypt depth ($p = 0.02$ and $p = 0.01$) in neonatal rabbits were elevated in all DA-supplemented treatments, while a linear and quadratic effect was observed in jejunum, but a quadratic effect was observed in duodenum ($p < 0.05$). The ileal villus height ($p < 0.01$) was also elevated in all DA-supplemented treatments exhibiting both linear and quadratic effects ($p < 0.05$). Moreover, the expression levels of occludin ($p = 0.04$), zonula occludens-1 (ZO-1) ($p < 0.01$), IGF-1 ($p = 0.03$), and solute carrier family 5 member 9 (SCL5A9) ($p < 0.01$) in jejunum were markedly elevated in the DA175 and 340 mg/kg DA-supplemented groups and were showing linear and quadratic effects ($p < 0.05$). Finally, inflammation-related gene expression levels such as NF-κB, TLR4, and MYD88 in jejunal ($p = 0.01$, $p = 0.04$, and $p < 0.01$) and ileal ($p = 0.04$, $p = 0.04$, and $p = 0.03$) epithelium were decreased in the DA170 and DA340 groups ($p < 0.05$), while linear and quadratic effects were observed for TLR4 in jejunum and for NF-κB and TLR4 MYD88 in ileum ($p < 0.05$). In summary, as a functional additive, maternal DA supplementation with 170 and 340 mg/kg DA during lactation can promote the growth of neonatal rabbits, which is related to improved antioxidative capacity and immunity, as well as improved intestinal health in neonatal rabbits.

**Keywords:** phytoestrogen; daidzein; intestinal health; growth performance; rabbit

## 1. Introduction

In recent decades, the reproductive performance of rabbits has been greatly improved by using artificial insemination and cycled production, as well as the application of prolific genetic strains [1]. However, increased fertility (for example, litter size) usually leads to a lack of maternal milk for neonatal rabbits, which subsequently increases their morbidity and mortality and subsequently causes economic loss for the rabbit industry [2]. In addition, it is universally known that oxidative stress and inflammation response are very common in animal production, especially for young animals [3–5]. Therefore, the avenue to improve the growth performance and health of neonatal rabbits has dramatically attracted worldwide research interest. Previous studies indicated that various hormones and growth factors are involved in the regulation of mammalian growth and metabolism processes [6–9]. For instance, hormones like progesterone (P), prolactin (PRL), and growth hormone (GH) play a crucial role in maintaining the development of mammary glands and improving milk yield both in cows and sows during the transition from gestation to lactation [10–13]. Moreover, it was reported that progesterone could cooperate with PRL in remodeling the mammary gland through the progesterone-dependent induction of mammary epithelial PRL receptors [14]. Additionally, GH can promote the growth of a wide variety of animal species via regulating the enzymatic activities, transporter functions, and expressions of critical metabolic genes [15].

Estrogen (E2) is one of the critical hormones that is responsible for the development of secondary sexual characteristics and plays a critical role in fertility and infertility in mammals [15,16]. It is well known that E2 exerts its role via binding to two subtypes of specific receptors (estrogen receptors, ERα and ERβ) located in target tissues. A previous study indicated that ERα activation can promote cell proliferation in a wide variety of tissues and organs, whereas ERβ activation is mainly involved in regulating cellular apoptosis [17]. Phytoestrogens are a kind of chemical compound that occurs naturally in plants and that have a similar chemical structure to endogen estradiol-17β (E-2) [18]. Importantly, phytoestrogens were found to have the same mechanisms of E-2 action [17,19]. A previous study also indicated that phytoestrogens have both ERs-dependent and independent mechanisms of action (e.g., they could directly activate receptors of insulin-like growth factor) [18].

Daidzein (DA), a kind of isoflavone, is one of the major phytoestrogens which is abundant in soybeans, pasture grasses, and whole cereals [20]. Previous studies indicated that dietary DA powder supplementation doses ranging from 50 to 200 mg/kg could promote growth, improve animal fertility, and ameliorate oxidative stress and inflammatory responses in rabbits, cows, and hens [20–23]. However, the DA supplementation of more than 400 mg/kg was observed to have a negative effect on body weight gain and splenic morphology in piglets [24]. Therefore, the effect of DA supplementation may be closely related to its different doses, sources, and animal species [25–27].

Numerous studies explored the positive effect of DA as a functional additive in improving growth performance, antioxidants, and anti-inflammation in cows, pigs, and humans [23,28,29], but few studies focused on rabbits. Moreover, this is the first study to explore the influence of maternal DA supplementation on growth and intestinal development in neonatal rabbits. Our previous study explored the influence of DA on the reproductive performance, immunity, and antioxidative ability of maternal does [20], and we hypothesized that maternal DA supplementation would improve the poor growth performance of neonatal rabbits due to increased litter size and insufficient milk yield. Hence, the objective of this experiment was to explore the influence and possible optimal dose of maternal DA powder supplementation during lactation on growth performance, antioxidative capacity, and intestinal health in neonatal rabbits. In addition, we also explored the effect of DA on the levels of hormones in maternal rabbits. Moreover, the potential mechanism behind the action of DA supplementation was also evaluated.

## 2. Materials and Methods

### 2.1. Animals, Housing, and Diets

A total of 152 five-month-old delivered New Zealand white does (maternal does received the same diet, management, synchronization of estrous, insemination, and delivery; 3.94 ± 0.05 kg) were randomly allocated to 4 groups with thirty-eight replicates of one doe each (one maternal doe/cage, n = 38). Experimental does were fed with a control diet (CD) containing 85 mg/kg (DA85), 170 mg/kg (DA170) and 340 mg/kg (DA340) DA. DA is a form of powder with a purity of 99.8% and was purchased from Guilin Fengpeng Bio. Co., Ltd. (Guilin, China). The experimental period lasted for 30 days. All maternal does were housed in separate cages (60 × 25 × 33), and equipment with a box for neonatal does, water, and feed was free to access. The room temperature was adjustable and was set between 15 and 25 °C during the experimental period. The experimental diets were formulated to meet the nutritional needs of rabbits according to recommendations [30]. The ingredients and chemical compositions are shown in Table 1, and the chemical analysis was previously described [20]. We recorded the litter size, weighed the individual body weight (BW), and calculated the average daily gain (ADG) of neonatal rabbits during the trial.

**Table 1.** Composition and chemical analysis of experimental diets.

| Item | %, as Fed |
|---|---|
| Alfalfa meal | 32.00 |
| Corn | 30.00 |
| Wheat bran | 20.00 |
| Rice bran | 9.81 |
| Soybean meal | 6.00 |
| Calcium bicarbonate | 1.00 |
| L−Lysine | 0.30 |
| L−Threonine | 0.10 |
| Tryptophan | 0.01 |
| Choline chloride | 0.15 |
| DL-methionine | 0.08 |
| Sodium chloride | 0.20 |
| Mineral-vitamin premix [1] | 0.35 |
| Chemical composition | |
| Digestible energy (MJ/kg) | 11.17 |
| Dry matter, % | 84.91 |
| Crude protein, % | 15.94 |
| Ether extract, % | 4.23 |
| Ash, % | 4.88 |
| Ca, % | 0.89 |
| Total phosphorus, % | 0.83 |

[1] Mineral and vitamin premix. Cu 3 mg; Zn 70 mg; Mn 2.5 mg; Fe 50 mg; Se 0.1 mg; I 0.5 mg. Vitamin A 12,000 IU; vitamin D 900 IU; vitamin E 40 IU; vitamin K 30.5 mg; vitamin B6 0.6 mg; vitamin B12 0.003 mg; vitamin B1 0.2 mg; vitamin B2 1.6 mg; folic acid 0.05 mg; nicotinic acid 3.0 mg.

### 2.2. Sample Collection

On day 21, 12 maternal and neonatal does from the 38 replicates in each treatment (the body weight of each selected rabbit was near the average BW of each group) were selected for blood sample collection. For maternal does, blood was collected via puncturing the ear vein. The neonatal rabbits were anaesthetized with anhydrous ether and blood samples were collected via the heart. All blood samples were processed as described previously [20]. After blood sampling, these neonatal does were sacrificed to collect the duodenal, jejunal, and ileal samples (the middle part of each intestinal segment was approximately 4 cm), fixed with 4% paraformaldehyde solution for intestinal morphology examination. The intestinal mucosa samples were obtained from the residual intestinal segment and then immediately frozen in liquid $N_2$ and stored at −80 °C for RNA analysis.

### 2.3. Serum Hormones and Metabolite Level Analysis

To evaluate the hormone and metabolite levels in the serum, the hormone-related indexes such as insulin-like growth factor-1 (IGF-1) and leptin (LEP) were determined. The concentration of IGF-1 and LEP were tested with ELISA kits (Jiangsu Meimian Industrial Co., Ltd., Yancheng, China). Moreover, the serum metabolites, such as high-density lipoprotein cholesterol (HDL-C), low-density lipoprotein cholesterol (LDL-C), and triglyceride (TG), were tested with a commercial kit (The Institute of Nanjing Jiancheng Bioengineering Institute, Nanjing, China). All procedures were referred to in previous studies [20].

### 2.4. Serum Immunity and Antioxidant Indices Analysis

The concentrations of immunoglobulin (IgA/G/M), interleukin1-β (IL-1β), interleukin-10 (IL-10), and tumor necrosis factor-α (TNF-α) were tested with ELISA kits (Jiangsu Meimian Industrial Co., Ltd., China). Antioxidant indices such as catalase (CAT), glutathione peroxidase (GSH-Px), malondialdehyde (MDA), and total antioxidative capability (T-AOC) were tested with commercial kits (Institute of Nanjing Jiancheng Bioengineering, China). All procedures were referred to in previous studies [20].

### 2.5. Intestinal Morphology Analysis

The preserved intestinal samples were embedded in paraffin wax, sectioned with microtome (HistoCore MULTICUT, Leica, Wetzlar, Germany), and stained with hematoxylin and eosin. The following steps were strictly referred to in previous studies [31]. The distance from the tip of the villi to the villus–crypt junction was defined as villus height, and the invaginated depth between adjacent villi was recorded as crypt depth. Image analysis software (Image-Pro Plus 6.0) and a light microscope (BX43, Olympus, Shinjuku City, Japan) were used to measure the 10 intact, well-oriented, crypt–villus units [32].

### 2.6. RNA Isolation, cDNA Synthesis, and q-PCR

The frozen mucosa samples (duodenum, jejunum, and ileum; about 0.1 g) were homogenized in 1 mL of RNAiso Plus (Takara Biotechnology Co., Ltd., Dalian, China). The extraction, quality, and reverse transcription of RNA were referred to in a previous study [20].

The expression level of the target gene in intestinal mucosa was quantified using q-PCR. The oligonucleotide primers sequences utilized in the present study were designed by Primer Premier 5.0 software (Palo Alto, CA, USA) and synthesized by Sangon Biotech (Shanghai, China). The details of these specific primers are listed in Table 2, and the procedure of qPCR was strictly performed referring to a previous study [20]. The mRNA expression levels of the four groups were calculated using the $2^{-\Delta\Delta Ct}$ method with β-actin as the housekeeping gene [33].

**Table 2.** Sequences of primers for genes related to growth, intestinal barrier, and inflammatory response.

| Item | Primer Sequence (5′–3′) | Annealing Temperature (°C) | Product Size (bp) | Gene Bank ID |
|---|---|---|---|---|
| β-actin | F: atgcagaaggagatcaccgc<br>R: cgtcatactcctgcttgctga | 60.18<br>60.13 | 154 | NM_001101683.1 |
| Zonula occludens-1, ZO-1 | F: tgcaaaaagtgaaccgcgag<br>R: tccgcctttccctcagaaac | 59.97<br>59.96 | 405 | XM_051822263.1 |
| Insulin-like growth factor-1, IGF-1 | F: catcctgtcctcctcgcatc<br>R: ccgtatcctgtgggcttgtt | 59.97<br>60.00 | 165 | NM_001082026.1 |
| Occludin | F: gcggcgtcggcagatt<br>R: gtgcatctcaccaccgtaca | 60.57<br>60.04 | 179 | XM_008262318.3 |

**Table 2.** *Cont.*

| Item | Primer Sequence (5′–3′) | Annealing Temperature (°C) | Product Size (bp) | Gene Bank ID |
|---|---|---|---|---|
| Claudin-1 | F: aaagatgcggatggctgtca<br>R: caaagtagggcacctcccag | 60.04<br>60.04 | 207 | NM_001089316.1 |
| Tumor necrosis factor, TNF-α | F: ggccctcaggaggaagagt<br>R: ggtttgctactacgtgggct | 60.31<br>60.04 | 124 | NM_001082263.1 |
| Interleukin, IL-1β | F: gacctgttctttgaggccga<br>R: ttctccagagccacaacgac | 59.97<br>59.97 | 158 | NM_001082201.1 |
| Interleukin, IL-10 | F: catcagggagcacgtgaact<br>R: ggctttgtagacgccttcct | 60.04<br>60.04 | 160 | NM_001082045.1 |
| Nuclear factor-κB, NF-KB | F: atttgcagacagaaggcgga<br>R: tgggggctttgctgtcatag | 59.96<br>60.03 | 207 | XM_051819545.1 |
| Toll-like receptor 4, TLR4 | F: gccttctcagcaggaacact<br>R: gtagggcttttctgagccgt | 59.96<br>60.04 | 87 | XM_008273277.3 |
| Myeloid differentiation factor 88, MYD88 | F: ctgcagagcaaggagtgtga<br>R: gggtccagaaccaggacttg | 59.97<br>59.96 | 181 | XM_002723869.4 |
| Solute carrier family 5, member 9, SLC5A9 | F: ttcctgctggccatcttctg<br>R: agtggaagtccttcagcacg | 60.03<br>59.97 | 166 | NM_001105687.1 |
| Solute carrier family 2, member 1, SLC2A1 | F: gctgccctggatgtcctatc<br>R: gaggtccagttggagaagcc | 59.96<br>60.04 | 162 | NM_001105687.1 |

*2.7. Statistical Analysis*

All data were subjected to a one-way ANOVA test for a completely randomized design using the GLM procedure of SPSS 20.0 (IBM SPSS 20.0 for Windows). To determine the dose-dependent effect of incremental levels of dietary daidzein, linear and quadratic effects due to the levels of DA were measured. The cage was the experimental unit for litter size and maternal serum parameters, while individual neonatal does were the experimental unit for individual body weight, AGD, serum parameters, intestinal morphology, and gene expression. A $p$-value < 0.05 was denoted statistically significant. All data were exhibited as means and separate SEM. Statistical differences among the groups were performed via Duncan's multiple comparison.

**3. Results**

*3.1. Growth Performance*

Table 3 shows the markedly improved individual weights of the neonatal rabbits at 21 d in the DA170 and DA340 groups ($p < 0.05$), showing linear and quadratic effects ($p < 0.05$). However, ADG was not affected by dietary DA, but linear and quadratic effects were observed ($p < 0.05$). Moreover, maternal DA supplementation significantly improved the individual weights of the neonatal rabbits at 31 d, and linear and quadratic effects were also observed ($p < 0.05$).

*3.2. Serum Parameters (Hormones, Immunity, Metabolites, and Antioxidants)*

The serum parameters of the maternal does were also determined and we found that dietary DA supplementation at all three levels significantly elevated the serum concentrations of progesterone and estrogen in the maternal does ($p < 0.01$). Serum IgA, IL-10, and GSH-Px concentrations significantly increased in the DA170 and DA340 groups ($p < 0.05$) but decreased the MDA content in the maternal does (Table 4). Here, dietary DA shows both linear and quadratic effects on P, IgA, IL-10, GSH-Px, and MDA in maternal does ($p < 0.05$). Interestingly, maternal DA supplementation significantly increased the serum concentrations of IGF-I, IgA, and IgG in neonatal rabbits with linear and quadratic effects ($p < 0.05$). The serum LDL-C level was lower in the maternal DA-supplemented group than

in the CON group at all three levels ($p < 0.05$). DA supplementation at 170 and 340 mg/kg significantly increased the serum concentrations of HDL-C and GSH-Px in neonatal rabbits, while linear and quadratic effects were only observed in GSH-Px (Table 5). In contrast, the serum concentration of CAT was high in the DA85 and DA170 groups when compared to the CON group, but it decreased the concentration of IL-1β in the neonatal rabbits ($p < 0.05$), where only CAT showed linear and quadratic effects ($p < 0.05$).

**Table 3.** Growth performance of neonatal rabbits from maternal supplementation with DA [1].

| Item | Treatments | | | | SEM | *p*-Value | | |
|---|---|---|---|---|---|---|---|---|
| | CON [2] | DA85 [3] | DA170 [4] | DA340 [5] | | ANOVA | Linear | Quadratic |
| Litter size at 21 d (n) | 8.03 | 9.22 | 9.50 | 8.26 | 0.34 | 0.31 | Ns | Ns |
| Individual body weight at 21 d (g) | 353.98 [c] | 386.28 [bc] | 404.68 [a] | 402.63 [a] | 8.56 | 0.03 | <0.01 | <0.05 |
| Average daily gain during 0–21 d (g/d) | 16.08 | 18.47 | 19.18 | 19.11 | 0.33 | 0.17 | <0.05 | <0.05 |
| Litter size at 31 d (n) | 7.77 | 8.42 | 9.00 | 8.02 | 0.28 | 0.15 | Ns | Ns |
| Individual body weight at 31 d (g) | 589.61 [b] | 674.82 [a] | 699.02 [a] | 671.88 [a] | 18.25 | <0.01 | <0.05 | <0.01 |
| Average daily gain during 21–31 d (g/d) | 24.03 | 28.97 | 29.69 | 27.53 | 0.75 | 0.07 | Ns | <0.01 |

Different superscript letters within a row denote significant difference ($p < 0.05$). Ns, not significant. [1] Results expressed in means and separate SEM (n = 12). [2] CON, control diet; [3,4,5] DA85, DA170, and DA340, control diet supplemented with 85, 170, and 340 mg/kg DA powder.

**Table 4.** Serum parameters in maternal does supplemented with DA [1].

| Item | Treatments | | | | SEM | *p*-Value | | |
|---|---|---|---|---|---|---|---|---|
| | CON [2] | DA85 [3] | DA170 [4] | DA340 [5] | | ANOVA | Linear | Quadratic |
| Progesterone (nmol/L) | 0.89 [b] | 1.20 [a] | 1.19 [a] | 1.43 [a] | 0.05 | <0.01 | <0.01 | 0.01 |
| Estrogen (pmol/L) | 155.82 | 235.95 | 214.50 | 197.00 | 8.80 | <0.01 | Ns | <0.05 |
| LDL-C (mmol/L) | 1.15 [a] | 0.22 [b] | 0.34 [b] | 0.18 [b] | 0.14 | 0.04 | <0.05 | <0.05 |
| HDL-C (mmol/L) | 0.33 [c] | 0.50 [bc] | 0.81 [a] | 0.49 [bc] | 0.05 | <0.01 | Ns | <0.05 |
| Immunoglobulins (μg/mL) | 86.65 [c] | 118.15 [bc] | 122.54 [a] | 134.13 [a] | 6.32 | 0.04 | <0.01 | <0.05 |
| Immunoglobulins (μg/mL) | 36.73 | 57.44 | 58.61 | 46.97 | 3.45 | 0.06 | Ns | <0.05 |
| Immunoglobulins (μg/mL) | 647.60 | 678.06 | 737.72 | 811.34 | 28.00 | 0.18 | <0.05 | Ns |
| TNF-α (pg/mL) | 882.35 | 838.31 | 743.17 | 786.66 | 27.65 | 0.31 | Ns | Ns |
| Interleukin-1β (ng/L) | 76.78 | 66.52 | 61.03 | 71.19 | 2.26 | 0.05 | Ns | <0.05 |
| Interleukin-10 (ng/L) | 20.84 [c] | 25.26 [bc] | 41.43 [a] | 30.31 [ab] | 2.17 | <0.01 | <0.01 | <0.01 |
| Catalase (U/mL) | 15.66 | 21.09 | 28.30 | 25.07 | 3.49 | 0.62 | Ns | Ns |
| Malonic dialdehyde (nmol/mL) | 78.78 [a] | 63.24 [ab] | 49.44 [c] | 44.26 [c] | 4.04 | 0.01 | <0.01 | 0.01 |
| Glutathione peroxidase (U/mL) | 35.30 | 48.71 | 52.44 | 53.97 | 2.33 | 0.01 | <0.01 | <0.01 |
| T-AOC (U/mL) | 13.54 | 26.93 | 28.86 | 21.84 | 2.49 | 0.10 | Ns | Ns |

Different superscript letters within a row denote significant differences ($p < 0.05$). Ns, not significant. [1] Results expressed in means and separate SEM (n = 12). [2] CON, control diet; [3,4,5] DA85, DA170, and DA340, control diet supplemented with 85, 170, and 340 mg/kg DA powder. HDL-C, high-density lipoprotein cholesterol; IGF-I, insulin-like growth factor-I; LDL-C, low-density lipoprotein cholesterol; T-AOC, total antioxidant capacity; TNF-α, tumor necrosis factor-α.

### 3.3. Intestinal Morphology

Table 6 and Figure 1 show that maternal DA supplementation at all three levels significantly increased the villus height both in the duodenum and jejunum, and a quadratic effect was also observed ($p < 0.05$). DA supplementation at all three levels decreased the crypt depth but significantly increased the ratio of V/C in the jejunum and showed linear and quadratic effects ($p < 0.01$). DA supplementation at 170 mg/kg also increased the V/C in the duodenum and increased the villus height in the ileum, while a quadratic effect on

V/C in duodenum was observed ($p < 0.05$) and linear and quadratic effects on villus height in ileum were found ($p < 0.05$).

**Table 5.** Serum parameters in neonatal rabbits from maternal supplementation with DA [1].

| Item | Treatments | | | | SEM | *p*-Value | | |
|---|---|---|---|---|---|---|---|---|
| | CON [2] | DA85 [3] | DA170 [4] | DA340 [5] | | ANOVA | Linear | Quadratic |
| IGF-I (µg/L) | 129.94 [b] | 184.87 [a] | 173.77 [a] | 188.31 [a] | 6.48 | <0.01 | <0.01 | <0.01 |
| Leptin (µg/L) | 32.97 | 35.64 | 33.11 | 33.11 | 0.69 | 0.50 | Ns | Ns |
| LDL-C (mmol/L) | 8.53 | 3.80 | 4.93 | 5.63 | 0.60 | 0.03 | Ns | Ns |
| HDL-C (mmol/L) | 0.30 | 0.78 | 0.81 | 0.87 | 0.12 | 0.04 | Ns | Ns |
| Immunoglobulin A (µg/mL) | 100.84 [b] | 128.08 [a] | 155.18 [a] | 127.23 [a] | 5.78 | <0.01 | <0.05 | <0.01 |
| Immunoglobulin M (µg/mL) | 51.41 | 57.98 | 60.39 | 56.62 | 1.67 | 0.26 | Ns | Ns |
| Immunoglobulin G (µg/mL) | 418.77 [b] | 522.91 [a] | 593.72 [a] | 584.41 [a] | 22.03 | 0.01 | <0.01 | <0.01 |
| TNF-α (pg/mL) | 899.03 [a] | 730.48 [c] | 773.61 [ab] | 708.73 [c] | 25.91 | 0.03 | <0.05 | <0.05 |
| Interleukin-1β (ng/L) | 88.50 | 56.81 | 69.27 | 71.03 | 3.97 | 0.03 | Ns | Ns |
| Interleukin-10 (ng/L) | 12.66 | 20.41 | 27.91 | 23.19 | 2.71 | 0.28 | Ns | Ns |
| Catalase (U/mL) | 17.95 [c] | 58.06 [a] | 66.37 [a] | 50.99 [bc] | 6.70 | 0.04 | <0.05 | <0.05 |
| Malonic dialdehyde (nmol/mL) | 203.91 | 143.38 | 90.41 | 148.75 | 14.45 | 0.03 | Ns | <0.05 |
| Glutathione peroxidase (U/mL) | 155.92 | 181.65 | 272.31 | 315.03 | 23.48 | 0.03 | <0.01 | <0.05 |
| T-AOC (U/mL) | 16.66 | 24.66 | 21.78 | 18.26 | 2.047 | 0.56 | Ns | NS |

Different superscript letters within a row denote significant difference ($p < 0.05$). [1] Results expressed in means and separate SEM (n = 12). Ns, not significant. [2] CON, control diet; [3,4,5] DA85, DA170, and DA340, control diet supplemented with 85, 170, and 340 mg/k DA powder. HDL-C, high-density lipoprotein cholesterol; IGF-I, insulin-like growth factor-I; LDL-C, low-density lipoprotein cholesterol; T-AOC, total antioxidant capacity.

**Table 6.** Intestinal morphology of neonatal rabbits from maternal supplementation with DA [1].

| Item | Treatments | | | | SEM | *p*-Value | | |
|---|---|---|---|---|---|---|---|---|
| | CON [2] | DA85 [3] | DA170 [4] | DA340 [5] | | ANOVA | Linear | Quadratic |
| Duodenum | | | | | | | | |
| Villus height (µm) | 556.39 [b] | 718.74 [a] | 695.19 [a] | 695.38 [a] | 22.28 | 0.04 | Ns | <0.05 |
| Crypt depth (µm) | 90.02 | 85.89 | 76.77 | 89.72 | 3.06 | 0.34 | Ns | Ns |
| V:C | 6.18 [c] | 8.37 [a] | 9.06 [a] | 7.75 [bc] | 0.38 | 0.02 | Ns | <0.01 |
| Jejunum | | | | | | | | |
| Villus height (µm) | 498.69 | 656.13 | 682.55 | 676.15 | 23.06 | <0.01 | <0.05 | <0.05 |
| Crypt depth (µm) | 100.11 [a] | 82.47 [b] | 73.06 [b] | 77.85 [b] | 3.11 | <0.01 | 0.01 | <0.01 |
| V:C | 4.98 [b] | 7.96 [a] | 9.21 [a] | 8.49 [a] | 0.45 | 0.01 | <0.01 | <0.01 |
| Ileum | | | | | | | | |
| Villus height (µm) | 401.10 [c] | 471.50 [bc] | 535.13 [a] | 501.18 [a] | 15.73 | <0.01 | <0.01 | <0.01 |
| Crypt depth (µm) | 68.00 | 64.62 | 66.84 | 63.72 | 4.21 | 0.86 | Ns | Ns |
| V:C | 5.90 | 7.30 | 8.01 | 7.86 | 0.62 | 0.40 | Ns | Ns |

Different superscript letters within a row denote significant difference ($p < 0.05$). [1] Results expressed in means and separate SEM (n = 12), Ns, not significant. V: C, villus height: crypt depth. [2] CON, control diet; [3,4,5] DA85, DA170, and DA340, control diet supplemented with 85, 170, and 340 mg/kg DA powder.

### 3.4. Intestinal Barrier Functions' Gene Expressions

As shown in Figure 2, the expression levels of occludin and ZO-1 in the jejunal epithelium were elevated in the DA170 and DA340 groups, while linear and quadratic effects on occludin, ZO-1, and claudin-1 were found, but no DA effect on claudin-1 was found ($p < 0.05$). DA supplementation at 340 mg/kg also elevated the expression level of Claudin-1 in ileum, but no linear or quadratic effect was observed ($p < 0.05$). Moreover, the expression levels of IGF-I in the jejunum and ileum and SLC5A9 in the jejunum were

elevated in the DA170 and DA340 groups ($p < 0.05$). IGF-I showed linear and quadratic effects in jejunum and ileum, while the same effects on SLC5A9 in jejunum were observed ($p < 0.05$). Maternal DA supplementation elevated the SLC2A1 expression level in ileum, but no linear or quadratic effect was found ($p < 0.05$).

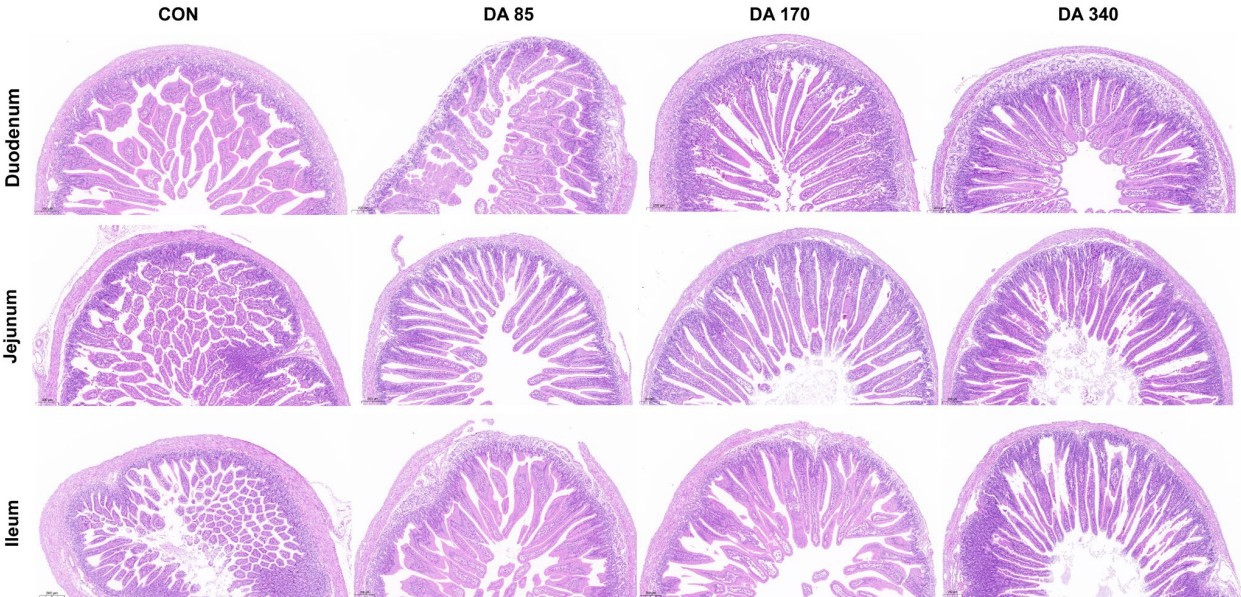

**Figure 1.** Intestinal morphology of neonatal rabbits from maternal supplementation with DA (H&E; ×50). CON, control diet; DA85, DA170, and DA340, control diet supplemented with 85, 170, and 340 mg/kg DA powder.

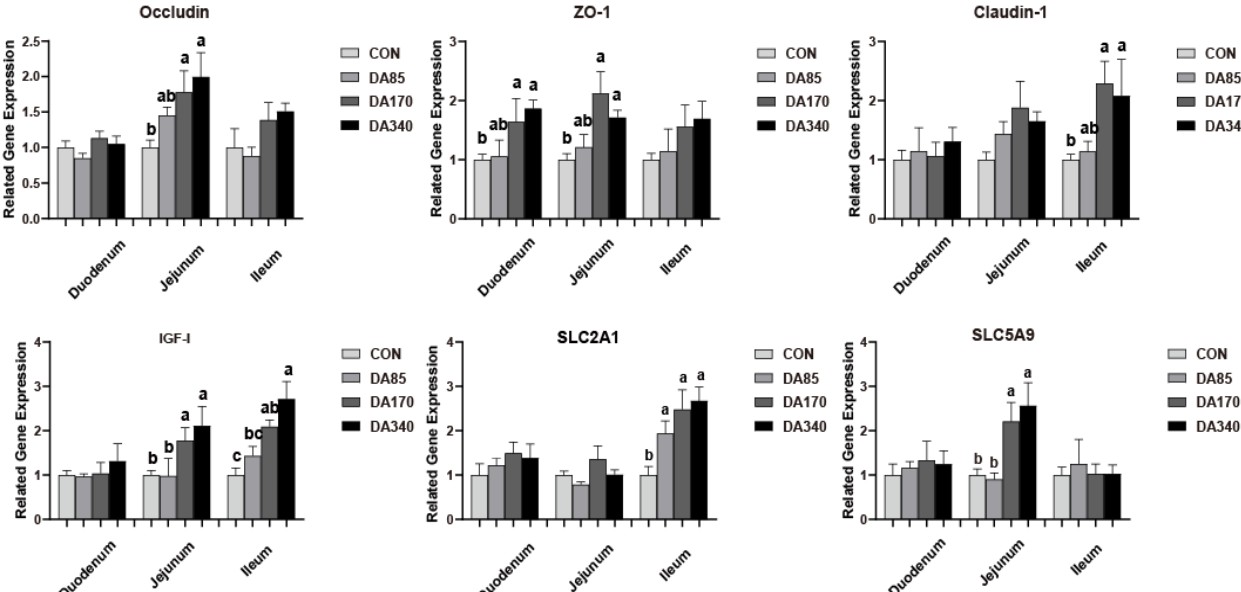

**Figure 2.** Critical gene expression related to intestinal barrier functions in neonatal rabbits from maternal supplementation with DA. CON, control diet; DA85, DA170, and DA340, control diet supplemented with 85, 170, and 340 mg/kg DA powder. IGF-I, insulin-like growth factor-I; SLC2A1, solute carrier family 2 (facilitated glucose transporter), member 1; SCL5A9, solute carrier family 5 (facilitated glucose transporter), member 9; ZO-1, zonula occludens-1. Different superscript letters within a column denote significant difference ($p < 0.05$).

### 3.5. Intestinal Inflammatory Responses' Gene Expressions

Figure 3 shows that maternal DA supplementation at all three levels reduced the expression level of NF-κB in the jejunal epithelium, where linear and quadratic effects were observed ($p < 0.05$). The expression levels of jejunal IL-1β and ileal NF-κB in the neonatal rabbits of the DA170 and DA340 groups were decreased, while NF-κB showed linear and quadratic effects ($p < 0.05$). DA supplementation at a higher dose (340 mg/kg) increased the IL-10 expression level in the intestinal epithelium, and linear and quadratic effects were only observed in the duodenum and jejunum ($p < 0.05$). Importantly, maternal DA supplementation markedly reduced the jejunal epithelium critical-inflammation-associated molecules' expression levels such as TNF-α and MYD88, while both of them did not show linear or quadratic effects ($p < 0.05$). Neonatal does in the DA170 and DA340 groups showed decreased expression levels of TLR4 both in the jejunum and ileum, and linear and quadratic effects were observed ($p < 0.05$). Moreover, neonatal does in the DA170 and DA340 groups showed decreased expression levels of NF-κB and MYD88 in the ileal epithelium, with linear and quadratic effects ($p < 0.05$).

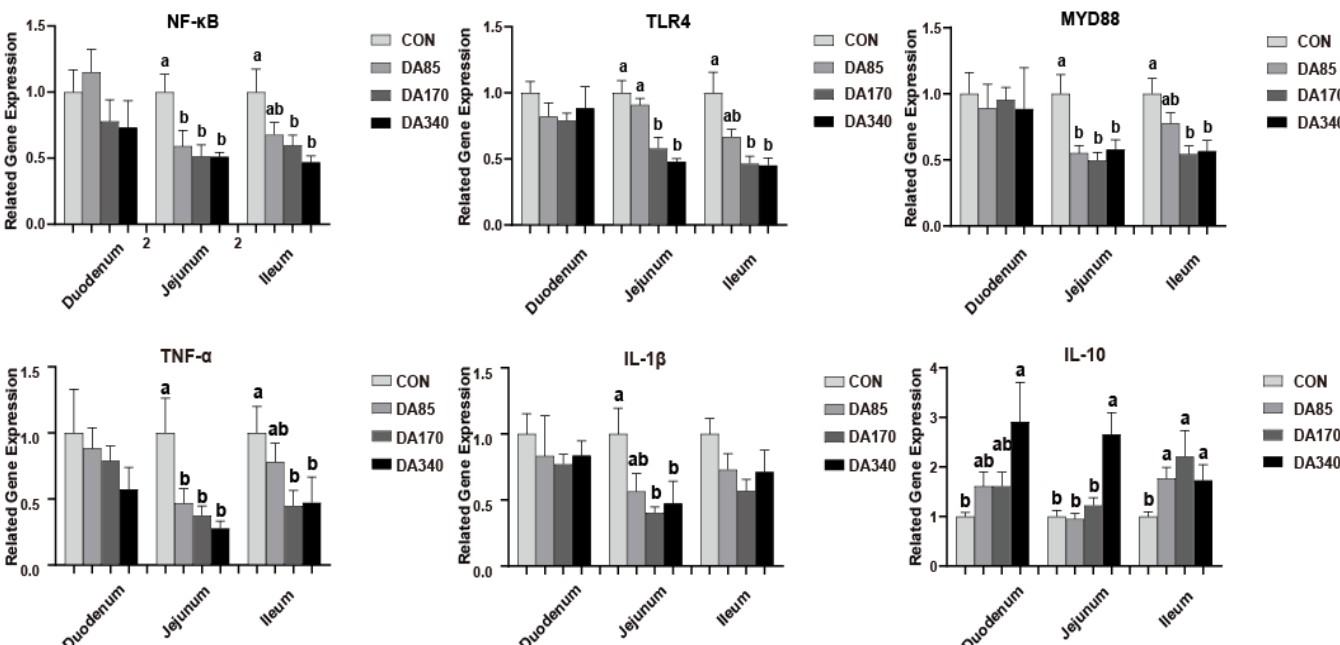

**Figure 3.** Critical gene expression related to inflammatory response in neonatal rabbits from maternal supplementation with DA. CON, control diet; DA85, DA170, and DA340, control diet supplemented with 85, 170, and 340 mg/kg DA powder. TLR4, Toll-like receptor 4; MYD88, myeloid differentiation factor 88; NF-κB, nuclear factor-κB. Different superscript letters within a column denote significant difference ($p < 0.05$).

## 4. Discussion

Mammalian growth and reproductive performance are regulated by various hormones and growth factors. Estrogen plays a critical role in nutrient metabolism and deposition, and it is responsible for regulating the female reproductive system [34–37]. As a critical phytoestrogen, DA was reported to improve the reproductive performance or growth performance in an array of animal species, including laying hens, sows, and rabbits [20,25,36]. Here, we evaluated the influence of maternal DA supplementation during lactation on the growth performance and intestinal health of neonatal rabbits. Our study shows that maternal DA supplementation markedly increased the individual weights of neonatal rabbits. This result was similar to the result of previous studies, in which DA supplementation markedly increased the litter weight and individual body weight of rats and piglets [27,38].

Insulin-like growth factors (IGFs) play a vital role in cell proliferation and growth processes by constituting a family of cellular modulators with their receptors and binding proteins [20]. This study showed that maternal dietary DA increased the IGF-1 concentration in the serum of neonatal rabbits, which may be associated with their improved growth performance during the suckling period [27,38]. Moreover, the elevated serum concentrations of estrogen and progesterone in the maternal does upon DA supplementation may also contribute to the elevated growth of the neonatal rabbits, as the two endocrine hormones play critical roles in regulating nutrient metabolism and maintaining lactation for animals during the lactating period [20,34,36,39]. LDL-C participates in the transportation of lipids in the body and has been recently looked at as one of the risk markers and the primary treatment target for hyperlipidemia and cardiovascular disease [40]. Here, DA supplementation reduced the serum LDL-C concentrations both in maternal does and neonatal rabbits, indicating a positive effect of DA in improving the lipid metabolism. Immunoglobulins such as IgG, IgM, and IgA play a vital role in cytoimmunity and humoral immunity and can serve as a critical index of immunity [41]. DA, a kind of phytoestrogen, could influence specific and nonspecific immunity by regulating the endocrine system in animals [42]. Results show that DA supplementation markedly increased the serum IgA and IgG concentrations in neonatal rabbits. The serum IgA and IgM concentrations were higher in 170 mg/kg DA-supplemented maternal does. Thus, we infer that neonatal does may acquire partial immunity from maternal does through suckling milk. These results are in accordance with previous reports that dietary DA supplementation could modulate the biological processes related to the specific and nonspecific immunity of mammalian animals [27,43,44]. Furthermore, dietary DA at 170 mg/kg markedly decreased the serum IL-1$\beta$ concentration both in the maternal and neonatal does, which suggested an anti-inflammatory role of DA. Similarly, DA supplementation also suppressed the secretion of proinflammatory cytokines like IL-6 or TNF-$\alpha$ in mice and sows [36,45]. And maternal DA supplementation at 340 mg/kg shows a significant reduced concentration of TNF-$\alpha$ in neonatal rabbits.

The over-production of ROS leads to the oxidation of lipids, proteins, DNA, and other tissue components, which subsequently causes tissue damage [46]. Antioxidant enzymes (e.g., GSH-Px) constitute the first line of defense against ROS [47]. Previous studies showed that DA can act as a more effective antioxidant than those conventionally used antioxidants like vitamin E [48,49]. It was reported that DA has two mechanisms in mediating antioxidant activity [29]. On the one hand, DA could directly scavenge radicals and bind to membranes, changing the membranous fluidity to impede the migration of radicals in liposomal membranes. On the other hand, DA exhibited an antioxidant effect indirectly by increasing the antioxidant enzyme activities in which CAT and GSH-Px were included. The results showed a significantly increased serum GSH-Px concentration and reduced MDA content in both the maternal does and neonatal rabbits of the DA170 group, which further proved the antioxidant nature of DA.

Intestinal nutrient digestion and absorption ability mainly depend on the integrity of the intestinal villus–crypt structure [50]. A previous study indicated that the mucosa architecture of neonates changes markedly during intensive remodeling, especially for villi density, shape, and size [51]. Here, maternal DA supplementation increased the villus height and the ratio of villus height to crypt depth (V/C) in the duodenum and jejunum of neonatal rabbits. Similar results were obtained in a previous study on juvenile turbot [52]. The improved development of intestinal morphology may be due in part to the elevated IGF-I concentration in the serum, as it was reported that IGF-I could promote cell proliferation and differentiation, especially for intestinal epithelium [53,54].

Importantly, the critical functional gene expression levels, such as the jejunal occludin, ZO-1, IGF-I, and SCL5A9, were elevated in the DA170 and DA340 groups of neonatal rabbits. The occludin and ZO-1 are two major components of tight junction proteins, which play a critical role in maintaining the integrity and permeability of intestinal epithelium [55]. However, the SLC5A9 can act as a critical transporter for glucose and other sugar units such

as D-Mannose, 1,5-anhydro-D-glucitol, and D-fructose in the intestinal epithelium [56–58]. Moreover, NF-κB is closely related to inflammation and could regulate the transcriptional activation of an array of target genes, including cytokines and proinflammatory mediators, whereas daidzein has been reported to inhibit NF-κB activation [26]. Maternal DA supplementation attenuated the intestinal inflammatory response of the neonatal rabbits, as indicated by decreases in the TNF-α and IL-1β expression levels as well as the downregulation of the MYD88, TLR4, and NF-κB expressions in the intestinal epithelium. The MYD88 triggers the TLR4/NF-κB signaling pathway, which subsequently regulates the inflammatory response by provoking the secretion of cytokines such as IL-1α, IL-1β, and TNF-α [59,60]. The anti-inflammatory effect of DA was also observed in previous studies on sows and Wistar rats [61–63].

## 5. Conclusions

The present study shows that maternal dietary DA at 170 and 340 mg/kg during lactation can effectively improve growth performance, immunity, and intestinal health in neonatal rabbits. The mechanisms of action might be closely associated with inhibiting inflammatory responses via decreased critical inflammatory signaling pathways' gene expression (NF-KB/TLR4/MYD88) and improved immunity (IgA and IgM) and antioxidative capacity (CAT and GSH-Px), as well as improved intestinal barrier (occludin/ZO-1/claudin-1) functions. Overall, from the point of view of economic efficiency, the optimum additive level of industrial application is 170 mg/kg.

**Author Contributions:** Conceptualization, J.H., methodology, H.X. and E.Y.; validation, H.X. and E.Y., histological, biochemical, q-PCR, and data analysis, H.X. and E.Y.; investigation, H.W., G.F. and H.S., resources, J.H., H.X. and B.J.; data curation, J.H. and B.J.; writing—original draft preparation, E.Y.; writing—review and editing, E.Y. and J.H.; project administration, J.H. and H.X.; funding acquisition, H.X. All authors have read and agreed to the published version of the manuscript.

**Funding:** This research was funded by the Shandong Industrial Technology System of Special Economic Animals (ADAIT-21-03), Shandong Scientific Research Program of College and University (No. J18KB067), and Weifang Scientific Research Program (No. 2019GX049), Shandong, China.

**Institutional Review Board Statement:** The experimental protocols (treatment, housing, husbandry, and slaughtering conditions) used in this study were granted by the Sichuan Agricultural University Animal Care and Use Committee, Chengdu, China (No. 20181105).

**Data Availability Statement:** The data used to support the findings of this study are available from the corresponding author upon request.

**Conflicts of Interest:** The authors declare no conflict of interest.

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
