# Peer review of "Maternal Daidzein Supplementation during Lactation Promotes Growth Performance, Immunity, and Intestinal Health in Neonatal Rabbits"

_agriculture, doi:10.3390/agriculture13091654_

Round 1

Reviewer 1 Report

A sloppy crafted rootless work. My evaluation report is below.

The title does not reflect the purpose of the study.

There are some flaws in the design of this study.

1-In fact, while there are two groups, a model is created as if there are 4 groups.

2 I think there would not be a serious difference if only those fed with DA and the control were analyzed. In such a case, ANOVA would not be used for analysis.

3-Although a model for oxidative stress has not been established, oxidative stress and inflammation have been evaluated.

 4-Although the main biomarker of inflammation is IL-6, it was not used in the study.

Although 5-Total antioxidant capacity was used, Total oxidant capacity was not used.

6-In my opinion, there is a lack of motivation here. In which case do the authors intend to use this DA nutritional therapy.

7-They evaluated the effects of this nutrition by measuring the villi length in the gastrointestinal tract and why did they not examine the liver. However, any response related to the gastrointestinal tract first occurs in the liver.

8-If bowel villi length is to be evaluated, Caspase-3 expression should be evaluated first.

9- I think that histological evaluation is not enough. The authors did not explain how they performed the histological assessment. They explained sloppily with only two references. When I examined the relevant reference, it was stated that the picture used a magnification unit of 40 and there is a size bar. However, in this study, it is stated which crypt depth and villi size are examined at 5 magnification and there is no size bar.

10-When I read the author contributions, it is as if the histological assessment was not done by the authors of this study.

Author Response

Dear Reviewer,

The point to point response to the comments, please see the attachment.

Reviewer 2 Report

Regarding MS entitled ‘’ Effect of maternal daidzein supplementation during lactation on growth performance, immunity, and intestinal health in neonatal rabbits’’

L23. Define treatment. The abbreviations should be defined at its first.

L65. More information should be added about Daidzein’s beneficial effects in rabbits from previous studies and try to explain the novelty of your study.

L79. Add hypothesis.

L86. Delete and

L88. How many replicates per treatment?

L89. Rephrase

L143. Does one internal control gene give accurate results? At least two internal control genes? How do the authors confirm that β-actin is a good candidate internal control gene?

Table S1. Are these sequences published previously or did the authors determine them? If the authors determine them, please add more details. Otherwise, add ref. to these sequences. The primer sequence of β-actin is missing?

Add a footnote and define the abbreviated genes.

L149. This is a dose-dependent study, the authors should perform orthogonal polynomial contrasts (linear and quadratic) to determine the effect of incremental levels of dietary daidzein on the measured parameters.

L150. Was doe the experimental unit for all parameters and even growth performance?

L165-166. Rephrase, grammar errors.

For all tables add means only and in a separate column add SEM, this is an old strategy to add mean ±SE or SD. Please define also your data are presented as means ± what as I guess. In addition, two columns for linear and quadratic p values.

Figures, please magnify the figures to be clear for the readers.

L289-295. The mechanism of action should be described.

There are many weak points in the discussion, the authors should deeply discuss their findings with previous studies and determine as possible the mechanism of action.

Moderate editing of English language required

Author Response

(The authors gave the same response as above.)

Reviewer 3 Report

Nice job. Please consider the following:

1. simple summary should be simple in explanation but not that short. Please add the importance of your research in it

2. Check grammar in line 86, 89, 

3. Could you explain how did you measure intestinal mofphology? What determines the villus and the crypt?

Minor errors that need to be modified are mentioned in comments to authors

Author Response

(The authors gave the same response as above.)

Reviewer 4 Report

Dear authors,

First of all, I would like to congratulate all the authors for their extraordinary work in this section of “Farm Animal Production”. The manuscript was well-written and the content was informative and well-presented. The manuscript will be a valuable contribution to this journal.

However, I’ve mentioned a few major comments that need to be addressed before the manuscript can be published: I also mentioned some minor corrections which need to be corrected in the comment section of the main manuscript file. Some of these include here:

Please follow the template of Agriculture MDPI journal template for your manuscript. 

Please follow the instructions of Agriculture MDPI Journal guidelines for your manuscript. In the Agriculture journal there is no need to write down the simple summary section separately, so please consider the journal instructions carefully. 

Line: 20: Please mention the replication, experimental doses of DA ?? groups, replication, and the total number of animals per treatment or in each replication.

Line: 21: Please rewrite it and try to clarify your statement, easy to understand, confusion about group classification and their dose level of DA.

Line 76: Please also indicate this point, is there any study gap, what's new in this? 

Line 86: Please mention their body weight, age, stage of lactation, and homogeneity of groups. 

Line 87: What about replications for each treatment group ?? Please mention replications for each treatment group.  

Please rewrite the conclusion part of this manuscript: To highlight the basic research gap which authors actually try to cover in this study along with their future recommendations, on the basis of their conclusion. Please write a conclusion with some strong evidence that can support your findings and try to make it more clear for the reader's interest with some future implementations and recommendations. Please elaborate a little bit more on your findings. 

Please set the entire list of references according to "Agriculture" MDPI journal instructions.  Please check the font style and make it correct according to the Agriculture Journal guidelines. Please carefully check all the references styles are strictly according to the Agriculture Journal references guidelines. Some references are too old so please try to use some recent studies.

Minor formatting and typographical errors.

Author Response

(The authors gave the same response as above.)

Round 2

Reviewer 1 Report

Every study should have a message to give. There are serious flaws in the design of this study. The study investigated oxidative stress in normal-born neonatal rabbits. And the goal of the study is to reduce oxidative stress in normal rabbits. In normal rabbits, everything is in a state of balance. Changing this balance will create a harmful situation. In normal neonatal rabbits, this study was done to change the oxidative stress in the equilibrium state, which leads to a disturbance of the balance of the whole body. Nothing can be better than normal. Therefore, I do not think that this study will contribute to the literature.

Author Response

Dear Reviewer:

Thank you for your precious suggestions, again. Our study titled "Maternal daidzein supplementation during lactation promotes growth performance, immunity, and intestinal health in neonatal rabbits". To our knowledge, neonatal rabbits are weaning (a shift from milk to feed was initiated) in succession around day 20. Weaning is an inevitable and stressful process for mammals, as it may affect the intestinal morphological structure and barrier function with altered food status and management methods and so on, resulting in reduced feed intake and growth retardation, and consequent economic losses (Gharib et al., 2018). Seriously, accumulated evidence showed that weaning stress could cause mammalian intestinal barrier dysfunction by inducing physiological inflammation and oxidative stress within a short time after weaning (Chen et al., 2022). Hence, as a functional feed additive, it is potentially necessary to evaluate the antioxidant activity of daidzein in neonatal rabbits.

Reviewer 2 Report

Dear authors 

Thank you for your revisions, the manuscript has been improved.

Minor editing 

Author Response

Dears Reviewer:

Thank you for your precious suggestions, again. English language of this manuscript has been revised.

Best wishes.

Reviewer 4 Report

Dear authors, 

I’ve mentioned a few major comments that need to be addressed before the manuscript can be published: I also mentioned some minor corrections which need to be corrected in the comment section of the main manuscript file. Some of these include here:

You have mentioned these two tables only within the text, but in the manuscript, these tables are missing. Please check it carefully where they are: 

Table 1. Ingredients of experimental diets and chemical composition (Missing entire table)

Table S1. Sequences of primers for genes related to growth, intestinal barrier, and inflammatory response. (Missing entire table)

Please set the entire list of references according to "Agriculture" MDPI journal instructions.  Please check the font style and make it correct according to the Agriculture Journal guidelines. 

Best wishes 

Author Response

Dear Reviewer:

Thank you for your precious suggestions, again. The point-by-point revisions to the comments are listed as follows:

(â… )We would like to submit these two tables as supplementary material at first, and now we added them in the revised manuscript.

(â…¡) We set the entire list of references according to "Agriculture" MDPI journal instructions.

(â…¢)We revised the manuscript based on the "peer-review-30729311.v1" and the journals guidelines

Best wishes